# Wheat Germ Drying with Different Time-Temperature Combinations in a Fluidized Bed Dryer

**Der-Sheng Chan [1,2]** and **Meng-I Kuo [1,3,*]**

1   PhD Program in Nutrition and Food Science, Fu Jen Catholic University, New Taipei City 24205, Taiwan; dschan@ms58.hinet.net
2   Department of Information Technology, Lee-Ming Institute of Technology, New Taipei City 243, Taiwan
3   Department of Food Science, Fu Jen Catholic University, New Taipei City 24205, Taiwan
*   Correspondence: 062998@mail.fju.edu.tw; Tel.: +886-2-2905-2019; Fax: +886-2-2209-3271

**Abstract:** The development of an effective drying performance of the fluidized bed dryer (FBD) is crucial to reduce drying costs. The objective of this study was to investigate the drying performance of wheat germ (WG) with different time-temperature combinations in the FBD. The WG was dried at different set temperatures of 80, 100 and 120 °C. The moisture content (MC) and water activity (WA) of WG were measured. A mathematical model was proposed to develop an optimal drying condition. The changes in the MC of WG during drying in the FBD could be divided into the decreased period, the dynamic equilibrium period and the increased period. The product temperature of 45 °C and WA of 0.3 of WG drying could be attained by different time-temperature combinations. The mathematical model, which was developed in conjunction with different time-temperature combinations, could predict the dehydration time and the condensation time of WG for optimization the drying conditions. The WG dehydration at the heating stage and the WG condensation at the cooling stage could also be evaluated by the dehydration flux and the condensation flux, respectively. The optimal drying performance of WG exists in a compromise between promoting dehydration and reducing condensation. Information obtained from the analysis of dehydration flux and condensation flux with experimental data and simulation gave the guidelines for performing an effective drying of WG in the FBD.

**Keywords:** wheat germ; fluidized bed drying; moisture content; dehydration; condensation; mathematical model

## 1. Introduction

Wheat germ (WG), a by-product from the wheat milling industry, is a natural source of bioactive compounds [1–3]. However, raw WG degrades rapidly due to the lipid oxidation by enzymes, as well as the high moisture content (MC) of about 15% and the high water activity (WA) about 0.7. Controlling the MC or WA of WG are the most common and important methods to prevent WG from spoilage [4] and extend its shelf-life. Therefore, developing a drying process with proper time-temperature combinations for WG stabilization without reducing its nutrition value is crucial.

The fluidized bed dryer (FBD) has been extensively used for the process of food drying [5,6] since it can effectively reduce the drying time and maintain the nutrition value of material. Fluidization provides a high interface area for heat and mass transfer between material and hot air, leading to a final product with uniform moisture distribution. However, the disadvantage of FBD is the high cost of equipment and maintenance. Many theoretical models were proposed to validate the drying

process of FBD [4,7–14]. Fluidized bed drying is a thermal process, where the agricultural materials are dehydrated at lower temperatures (<100 °C) [4,12,15] or higher temperatures of 120–240 °C [4–6,16,17] by contacting with the convective hot air. Heating temperature and heating time are significant factors affecting the amount of MC inside WG during drying. Gili et al. [4] revealed that higher absolute drying rates were obtained as the air temperature increased from 90 °C to 150 °C. Moreover, from a commercial point of view, the thermal treatment with high temperature and prolonged heating time provides dried WG with a desired cooked flavor and a light golden-yellow or brown color [17]. However, at higher heating temperatures, nutritional loss and thermal input are higher during drying. In our previous study [18], the fluidized bed drying of WG at 80 °C was divided into a heating stage and a cooling stage. The increase of MC during WG drying at the cooling stage was observed due to the condensation of water vapor on the WG surface.

It is important to further understand the effects of different time-temperature combinations on the MC and WA of WG during FBD drying. The objective of this study was to investigate the drying performance of WG with different time-temperature combinations in the FBD. Additionally, a mathematical model was proposed to optimize the drying conditions.

## 2. Materials and Methods

### 2.1. Materials

The raw WG was purchased from the local supplier (Texture Maker Enterprise Co., Ltd.) in Taiwan and stored at −20 °C. Before the fluidized drying experiment, the WG sample was placed in an environment-controlled storeroom at 25 °C for 12 h.

### 2.2. Analytical Methods

The WA and MC of WG during the drying process in the FBD were measured. The WA of WG was acquired by the thermal hygrometer (Testo 635, Testo Inc., Lenzkirch, Germary). The MC of WG was performed on the basis of the AACC Method 44-19 [19]. 2 g ± 1 mg of WG sample was dried at 135 °C for 2 h.

### 2.3. Fluidized Bed Dryer and Process

The geometry and dimension of the FBD used in this study is shown in Figure 1. The drying equipment includes an air compressor (Shia Machinery Industrial Co. Ltd., Taichung City, Taiwan), heater (Shia Machinery Industrial Co. Ltd., Taichung City, Taiwan), inbuilt program logical controller (Shia Machinery Industrial Co. Ltd., Taichung City, Taiwan), sample bin, drying chamber, filter bags and outlet air motor (Shia Machinery Industrial Co. Ltd., Taichung City, Taiwan). The diameter of the porous mesh holder at the bottom of sample bin was smaller than the one used in our previous study [18] in order to improve the fluidized efficiency. The surrounding air with a temperature of 30 °C and the relative humidity of 75% was forced into the heater by air compressor and blew through the filter bags after passing the sample bin and the drying chamber. The inlet air temperature was measured by a K-type thermometer (Tecpel 318, Tecpel Co., Ltd., Taipei, Taiwan). The air velocity was measured by a gas flow transmitter (LABO-FG, GHM Messtechnik GmbH, Germany). The 2 kg particles of raw WG were fluidized with 1.2 m/s air velocity from the bottom of the sample bin in the FBD.

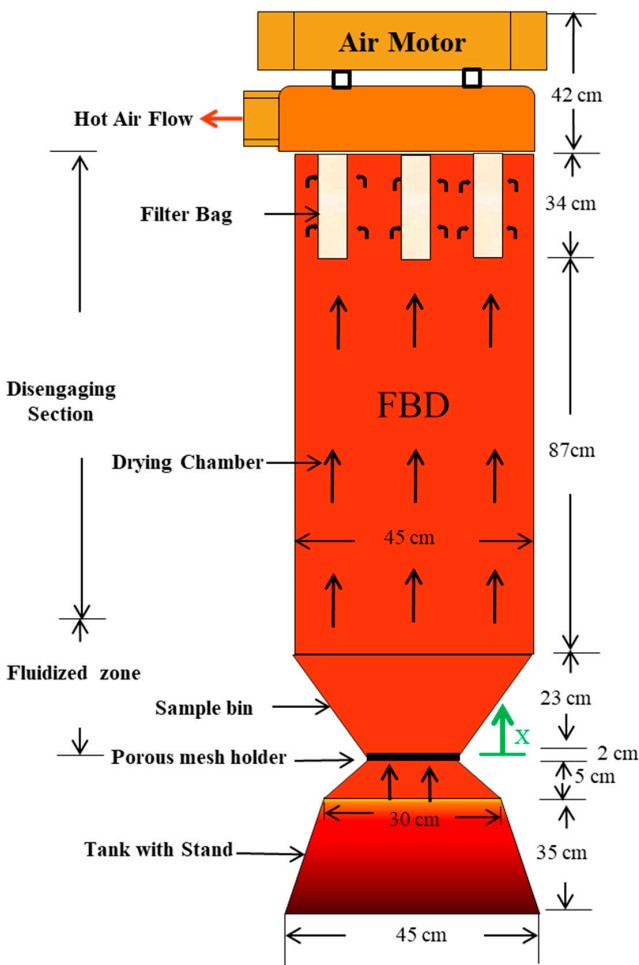

**Figure 1.** Geometry and dimension of the fluidized bed dryer (FBD) used in this study.

## 2.4. Drying Strategy

In this study, three different time-temperature combinations were studied. The entire process was divided into four stages: preheating, sample loading, heating and cooling (Table 1). The FBD was preheated at different set temperatures for 10 min. The 2 kg particles of raw WG were spread on the sample bin as a 7 cm thin layer within 1 min. After the WG sample was loaded, it was dried at 80 °C for 9.4 min, 100 °C for 7.5 min or 120 °C for 3.0 min, respectively. In this study, different time-temperature combinations were used for WG drying because the target WA of the final product was 0.3 ± 0.1, for better storage stability [20,21]. In order to obtain a final product temperature of around 45 °C, the cooling time of set temperatures of 80 °C, 100 °C and 120 °C was 14.6 min, 16.5 min and 21.0 min, respectively.

**Table 1.** Time allocation of each stage for wheat germ (WG) drying at 80–120 °C in the fluidized bed dryer.

| Stages (min) | Set Temperature | | |
| --- | --- | --- | --- |
| | 80 °C | 100 °C | 120 °C |
| Preheating | 10.0 | 10.0 | 10.0 |
| Sample Loading | 1.0 | 1.0 | 1.0 |
| Heating | 9.4 | 7.5 | 3.0 |
| Cooling | 14.6 | 16.5 | 21.0 |

## 3. Mathematical Modelling

### *3.1. Governing Equations*

For a detailed description of the mathematical model for the WG drying process in the FBD, including thermophysical properties and assumptions, we refer to our previous study [18]. The simple descriptions of various heat and mass transfer equations with initial and boundary conditions for WG drying process are as follows:

3.1.1. Microscopic Energy Balance for Wheat Germ

$$\rho_g C_{P,g} \frac{\partial T_g}{\partial t} = \frac{1}{r^2} \frac{\partial}{\partial r} (r^2 k_g \frac{\partial T_g}{\partial r}) \tag{1}$$

Initial condition:

$$T_g = T_{gi} \text{ for } 0 \leq r \leq r_g \tag{2}$$

Boundary conditions:

$$\frac{\partial T_g}{\partial r} = 0 \text{ for } r = 0 \tag{3}$$

$$k_g \frac{\partial T_g}{\partial r} = h(T_e - T_g) + \lambda D_e \frac{\partial C_{mg}}{\partial r} \text{ for } r = r_g \tag{4}$$

3.1.2. Microscopic Mass Balance for Wheat Germ

$$\frac{\partial C_{mg}}{\partial t} = \frac{1}{r^2} \frac{\partial}{\partial r} (r^2 D_e \frac{\partial C_{mg}}{\partial r}) \tag{5}$$

Initial condition:

$$C_{mg} = \frac{W_L X_{wi}}{M_w V_T} \text{ for } 0 \leq r \leq r_g \tag{6}$$

Boundary conditions:

$$\frac{\partial C_{mg}}{\partial r} = 0 \text{ for } r = 0 \tag{7}$$

$$- D_e \frac{\partial C_{mg}}{\partial r} = K_{de}(C_{mg} - C_e)f_{de} - K_{con}(C_{me} - C_{sat})f_{con} \text{ for } r = r_g \tag{8}$$

3.1.3. Macroscopic Energy Balance of Emulsion Phase

$$\rho_{em} C_{P,e} (\frac{\partial T_e}{\partial t} + u \frac{\partial T_e}{\partial x}) = \frac{\partial}{\partial x} (k_{em} \frac{\partial T_e}{\partial x}) - A_g(1 - \varphi_b)(k_e \frac{\partial T_g}{\partial r}) \tag{9}$$

Initial condition:

$$T_e = T_i \text{ for } 0 \leq x \leq H_f \tag{10}$$

Boundary conditions:

$$T_e = T_{in} \text{ for } x = 0 \tag{11}$$

$$\frac{\partial T_e}{\partial x} = 0 \text{ for } x = H_f \tag{12}$$

3.1.4. Macroscopic Mass Balance of Emulsion Phase

$$\frac{\partial C_{me}}{\partial t} = \frac{\partial}{\partial x} (D_v \frac{\partial C_{me}}{\partial x}) - A_g(1 - \varphi_b)(D_e \frac{\partial C_{mg}}{\partial r}) - R_{dl} \tag{13}$$

Initial condition:

$$C_{me} = \frac{RH_i P_{sat}(T_i)}{R_T T_i} \text{ for } 0 \le x \le H_f \tag{14}$$

Boundary conditions:

$$C_{me} = \frac{RH_{in} P_{sat}(T_{in})}{R_T T_{in}} \text{ for } x = 0 \tag{15}$$

$$\frac{\partial C_{me}}{\partial x} = 0 \text{ for } x = H_f \tag{16}$$

### 3.1.5. Macroscopic Mass Balance of Bubble Phase

$$\frac{\partial C_{mb}}{\partial t} + u \frac{\partial C_{mb}}{\partial x} = \frac{\partial}{\partial x}\left(D_v \frac{\partial C_{mb}}{\partial x}\right) + R_{dl} \tag{17}$$

Initial condition:

$$C_{mb} = \frac{RH_i P_{sat}(T_i)}{R_T T_i} \text{ for } 0 \le x\, H_f \tag{18}$$

Boundary conditions:

$$C_{mb} = \frac{RH_{in} P_{sat}(T_{in})}{R_T T_{in}} \text{ for } x = 0 \tag{19}$$

$$\frac{\partial C_{mb}}{\partial x} = 0 \text{ for } x = H_f \tag{20}$$

The variable equations listed in Table 2 are important and were used for the model simulation in this study. The variable equations, discussed in the Results and Discussion section, were described in detail in the Sections 3.2–3.4.

**Table 2.** Variables equations used in this study.

| Variable | Meaning | Expression |
|---|---|---|
| $R_{dl}$ | Rate of mass transfer | $K_{be}(C_{me} - C_{mb})$ |
| $P_{sat}(T)$ | Saturated vapor pressure | $0.1 \exp\left(27.0214 - \frac{6887}{T} - 5.31 \ln\left(\frac{T}{273.15}\right)\right)$ |
| $C_{sat}$ | Saturated concentration | $\frac{P_{sat}(T)}{R_T T}$ |
| $f_{step}$ | Step function | $\begin{cases} 1 & \text{if } t < t_h \\ 0 & \text{if } t \ge t_h \end{cases}$ |
| $f_{con}$ | Step function | $\begin{cases} 0 & \text{if } T_g < T_e \\ 1 & \text{if } T_g \ge T_e \end{cases}$ |

### 3.2. Simulation of the Inlet Air Temperature of FBD

The equations used to simulate the inlet air temperature of FBD ($T_{heater}$) from our previous study [18] were modified due to a higher environmental temperature of 30 °C and a higher environmental relative humidity of 75%:

$$T_{heater} = S_h \times f_{step} + S_c \times (1 - f_{step}) \tag{21}$$

$$S_h = T_{start} + (T_s - T_{start}) \times (1.0 - \exp(-2.0t/20)) \tag{22}$$

$$S_c = 319.15 + 1.01 \times (T_s - 319.15) \times \exp(-0.4 \times (t - t_h)/98.0) \tag{23}$$

where $S_h$ is the temperature at the heating stage; $S_c$ is the temperature at the cooling stage; $T_{start}$ is the measured temperature at the sample loading stage; $t_h$ is the heating time; $T_s$ is the set temperature (80, 100 and 120 °C).

### 3.3. Diffusivity of Moisture in Wheat Germ

The diffusivity of moisture inside WG is a function of temperatures and can be expressed by the Clausius-Clapeyron equation [12]:

$$D_e = D_o \exp(-\frac{E_a}{R_g T_g}) \tag{24}$$

### 3.4. Thermal Input for Wheat Germ Drying

The thermal efficiency of WG drying depends on the heating temperature and heating time. In our previous study [18], the thermal input (*TI*) was used to evaluate the time-temperature curve of WG:

$$TI = \int_0^t T_g dt \tag{25}$$

## 4. Results and Discussion

### 4.1. Measured and Simulated Responses of the Inlet Temperature of FBD

The experimental data and the predicted values of the heater temperature during drying at different set temperatures in the FBD are shown in Figure 2. The simulated results were in good agreement with the measured data during drying at different set temperature. The equations for heater temperature prediction could be employed to simulate the inlet boundary condition of the whole drying process. Furthermore, the cooling time required to drop the temperature from 80, 100 and 120 °C to 45 °C was 14.6 min, 16.5 min and 21.0 min, respectively. The above result indicates that the higher the set temperature of WG drying, the longer the cooling time.

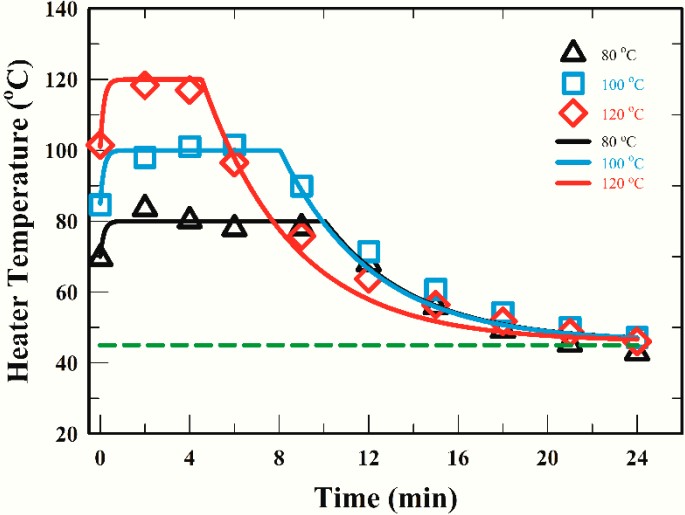

**Figure 2.** Profiles of the inlet air temperature (the heater temperature) during wheat germ drying in the FBD at different set temperature. The symbols represent the experimental data; the solid lines represent the simulated values; the dashed line represents the product temperature of 45 °C.

### 4.2. Model Verification

The mass and heat transfer equations were solved simultaneously by using COMSOL Multiphysics software version 5.1 (COMSOL Multiphysics, Co, Stockholm, Sweden) with a finite element method. The model parameters for simulation are given in Table 3. The model was validated by using the absolute average deviation (AAD) [4]. The values of AAD were 0.40, 0.47 and 0.26 for three set temperatures of 80, 100 and 120 °C, respectively (Appendix A). The above results indicated that the parameters (Table 3) selected to predict the MC of WG were adequate.

**Table 3.** Input parameter values.

| Panel A: Parameters and corresponding values used in this study | | | | | | | |
|---|---|---|---|---|---|---|---|
| Par. | Value | Par. | Value | Par. | Value | Par. | Value |
| $A_g$ | 3239 1/m [e] | $T_{gi}$ | 298.15 K [m] | $X_{wi}$ | 0.15 d.b. [m] | $h$ | 32.5 W/m$^2$ ·K [s] |
| $r_g$ | $9.26 \times 10^{-3}$ m [e] | $T_{in}$ | 303.15 K [m] | $\varphi_b$ | 0.92 (-) [m] | $K_{be}$ | $1.2 \times 10^{-3}$ m/s [s] |
| $H_f$ | 0.57 m [m] | $W_L$ | 2.0 kg [m] | $D_o$ | $7.5 \times 10^{-4}$ m$^2$/s [s] | $M_w$ | 0.018 kg/mol [t] |
| $RH_i$ | 0.75 (-) [m] | $u$ | 1.2 m/s [m] | $D_v$ | $1.0 \times 10^{-5}$ m$^2$/s [s] | $R_T$ | 0.08205 atm/(mol·K) [t] |
| $RH_{in}$ | 0.75 (-) [m] | $V_T$ | 0.214 m$^3$ [m] | $E_a$ | $29.4 \times 10^3$ J/mol [s] | $R_g$ | 8.314 J/(mol·K) [t] |

| Panel B: Parameters and corresponding values for different set temperature operation | | | |
|---|---|---|---|
| Parameter | Set Temperature | | |
| | 80 °C | 100 °C | 120 °C |
| $K_{con}$ (m/s) | $1.5 \times 10^{-3}$ [s] | $2.5 \times 10^{-3}$ [s] | $4.0 \times 10^{-3}$ [s] |
| $K_{de}$ (m/s) | $7.0 \times 10^{-3}$ [s] | $9.0 \times 10^{-3}$ [s] | $12.0 \times 10^{-3}$ [s] |
| n | 0.36 [s] | 0.37 [s] | 0.39 [s] |
| $T_{start}$ | 70.0 [m] | 85.0 [m] | 101.0 [m] |

[e]: estimated; [m]: measured; [s]: set; [t]: theoretical; Par.: parameter.

### 4.3. Measured and Simulated Responses of the Moisture Content of WG

The measured data and the simulated curves of the MC of WG during drying at 80, 100 and 120 °C are illustrated in Figure 3. The changes of the MC of WG during drying in the FBD could be divided into three different periods: the decreased period (0–8 min), the dynamic equilibrium period (8–18 min) and the increased period (18–24 min). The time of period depended on the time-temperature history of WG drying. An increase of MC of WG during the cooling stage was observed and was in accordance with our previous report [18]. Without the cooling stage, only the abrupt decreased period and the equilibrium period were found [4]. Moreover, the decrease of the MC of WG was the function of heating time and set temperature during drying. The MC values predicted were in good agreement with the experimental data, indicating that the model can be applied to evaluate the MC of WG during drying at both heating and cooling stages.

Moreover, the MC of WG decreased to a constant value at the end of the heating stage at 80 °C and 100 °C. However, when the WG was dried at 120 °C, the MC of WG continuously decreased at the end of heating stage and reached a constant value at the cooling stage, indicating a prolonging effect with higher temperature and short heating time.

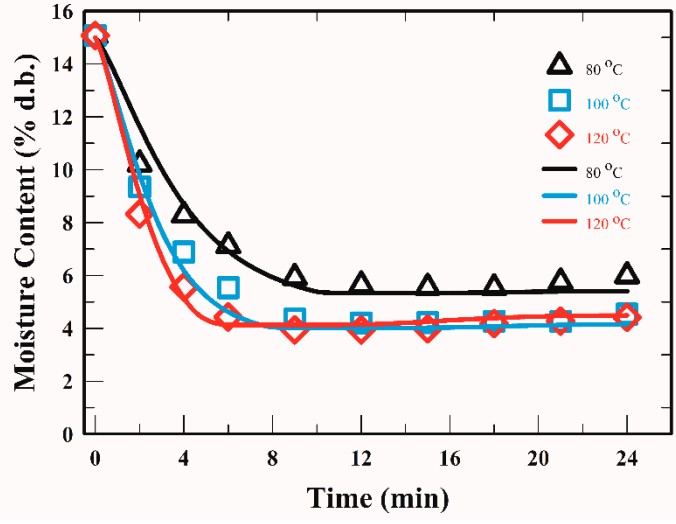

**Figure 3.** Profiles of the moisture content of wheat germ during drying in the FBD at different set temperature. The symbols represent the experimental data; the lines represent the simulated values.

*4.4. Measured and Simulated Responses of the Water Activity of WG*

A simple linear relationship between the WA and MC of WG was used as follows:

$$WA = 0.045 \times MC + 0.07 \tag{26}$$

The measured data and the fitting curves (Equation (26)) of the WA of WG during drying at different set temperatures are illustrated in Figure 4. The predicted values were consistent with the experimental measurement. It was observed that the WA of WG was very close to 0.3 when the WG was dried at 80 °C for 9.4 min. The WA of WG was slightly less than 0.3 when the WG was dried at the temperature of 100 °C for 7.5 min and 120 °C for 3 min. Furthermore, the WA of WG decreased with heating time and set temperature.

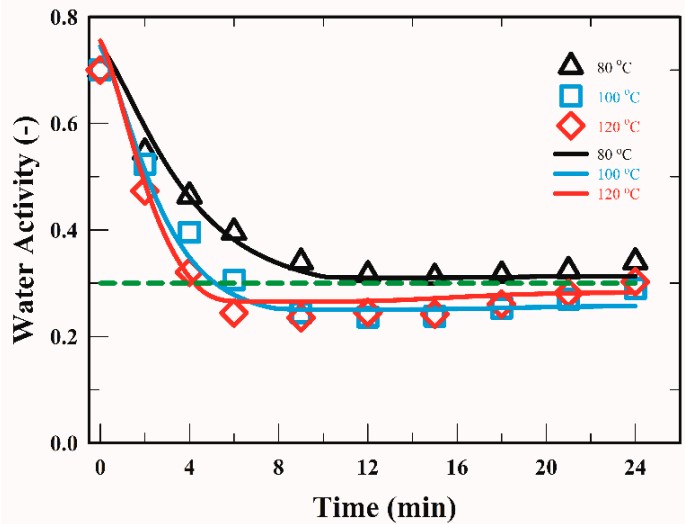

**Figure 4.** Profiles of the water activity of wheat germ during drying in the FBD at different set temperatures. The symbols represent the experimental data; the solid lines represent the simulated values; the dashed line represents the target water activity of 0.3.

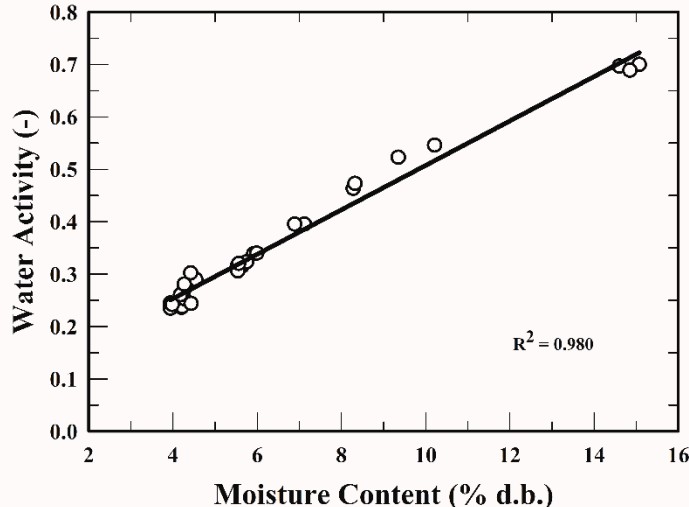

**Figure 5.** The relationship between the water activity and moisture content of wheat germ. The symbols represent the experimental data; the solid line is the regression line.

Both MC and WA were important parameters to evaluate food stability [20,21]. It only takes 30 min to measure the WA of food by using the thermal hygrometer. However, it takes at least 2 h to measure the MC of WG by using the AACC method. The above equation (Equation (26)) could be

used to predict the MC of WG once the WA was obtained. Figure 5 shows the correlation between the WA and MC of WG which can be considered as the mass equilibrium between the liquid phase and vapor phase.

### 4.5. Simulated Responses of the Air Temperature and Germ Temperature in the FBD Chamber

Changes in the values of the air temperature and the WG temperature during drying in the FBD are shown in Figure 6. The air temperature in the FBD rapidly dropped from preheating temperature after WG loading, after which the air temperature was increased because the heat transfer occurred by forced convection from heater to air. This result revealed that the air temperature in the FBD chamber during WG drying strongly depended on the heater temperature (Figure 2). Additionally, a heat transfer between air and WG by forced convection existed, resulting in the corresponding increase in the WG temperature. The air temperature and WG temperature gradually decreased during cooling stage. Both temperatures approached 45 °C at the end of cooling stage (Figure 6).

According to Figure 6, WG drying could be divided into three periods: the heating period, the dynamic equilibrium period (both air temperature and WG temperature were equal) and the cooling period. The time of heating period ranged from 0 to 5.5 min; that of dynamic equilibrium period from 5.5 to 8.3 min; and that of cooling period started from 8.3 min to the end of drying when the WG was dried at 80 °C and 100 °C in the FBD. Since the time of the heating stage was short (3 min), the air temperature would not be able to reach 120 °C before entering the cooling stage. No dynamic equilibrium period was found when the set temperature was 120 °C. Based on the above results, the set temperature and heating time strongly affected the temperature history of WG.

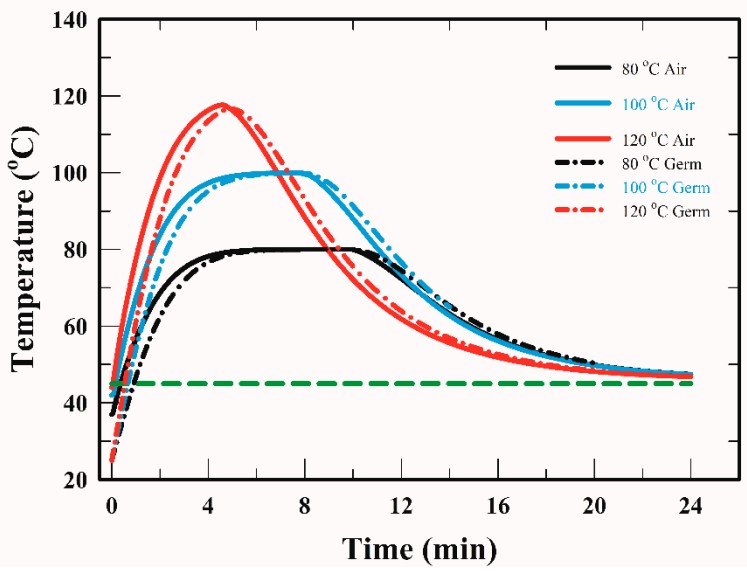

**Figure 6.** Simulated profiles of the air temperature and WG temperature during drying in the FBD at different set temperature. Solid lines represent the air temperature; dash-dot lines represent the WG temperature; the dashed line represents the product temperature of 45 °C.

### 4.6. Simulated Responses of the Thermal Input and the Diffusivity of Moisture in WG

The time-temperature curve of WG during drying was used to evaluate the thermal input (*TI*) in this study. The simulated thermal input of heating WG in the FBD at different set temperature is shown in Figure 7. During the heating stage, the *TI* of WG drying at 80 °C for 9.4 min was $3.9 \times 10^4$ K·min; that of WG drying at 100 °C for 7.5 min was $3.7 \times 10^4$ K·min; and that of WG drying at 120 °C for 3 min was $1.3 \times 10^4$ K·min. The performance of WG drying at 120 °C for 3 min was more efficient than that of WG drying at lower temperature. The above results indicate that the drying performance depends on the time-temperature combination.

The dehydration of WG was directly related to the moisture diffusion. The simulation results of the diffusivity of moisture inside the WG during drying at different set temperature in the FBD are shown in Figure 8. Results showed that the diffusivity of moisture in WG increased and then decreased at the heating stage and the cooling stage, respectively. The moisture diffusivity was in the order of 120 °C > 100 °C > 80 °C, as far as the heating stage was concerned. The diffusivity of moisture in WG during drying was related to the WG temperature (Figure 6).

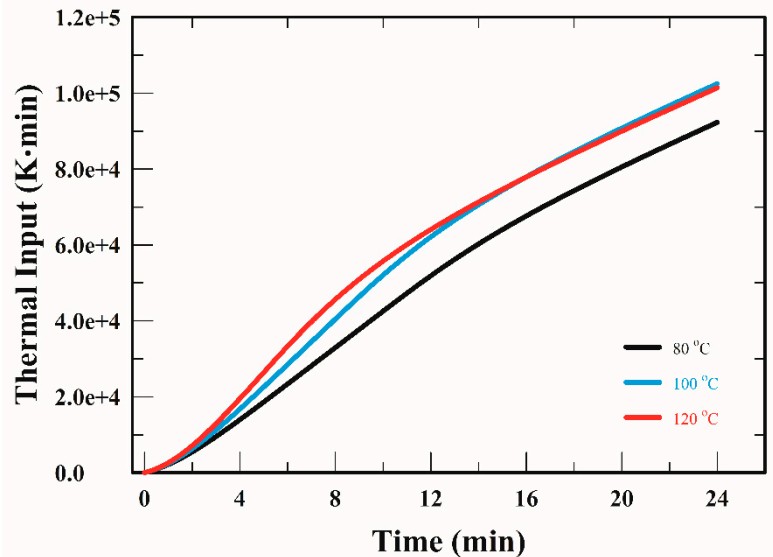

**Figure 7.** Simulated thermal input of heating germ in the FBD at different set temperature. 80 °C (black line), 100 °C (blue line) and 120 °C (red line).

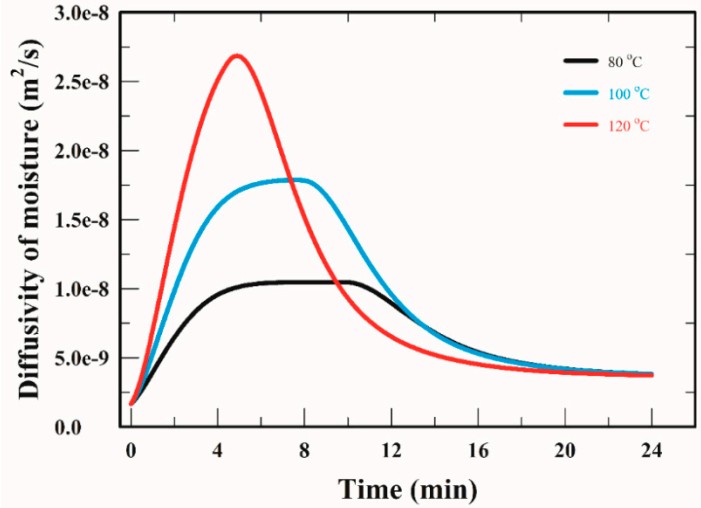

**Figure 8.** Simulated profiles of the diffusivity of water vapor inside WG during drying in the FBD. 80 °C (black line), 100 °C (blue line) and 120 °C (red line).

### 4.7. Simulated Responses of WG Temperature Distribution

The simulated temperatures of the WG center and WG surface during drying at different set temperatures in the FBD are shown in Figure 9A. During the heating stage, the temperature of the WG surface was higher than that of the WG center. However, the trend was opposite during the cooling stage. Based on the WG temperature, WG drying at 80 and 100 °C could also be divided into three periods: the increased period (0 to 5.0 min), the dynamic equilibrium period (5.0 to 9.0 min) and the

decreased period (more than 9.0 min). However, no equilibrium period was observed when the set temperature was 120 °C.

The temperature difference between the center and surface of WG during drying at different set temperatures in the FBD are shown in Figure 9B. After the sample loading stage, the value of the temperature difference rose sharply, and then, the value rapidly dropped during the heating stage. The value of temperature difference was negative at the final stage of heating and the initial stage of cooling. Finally, the value gradually reached zero at the final stage of cooling.

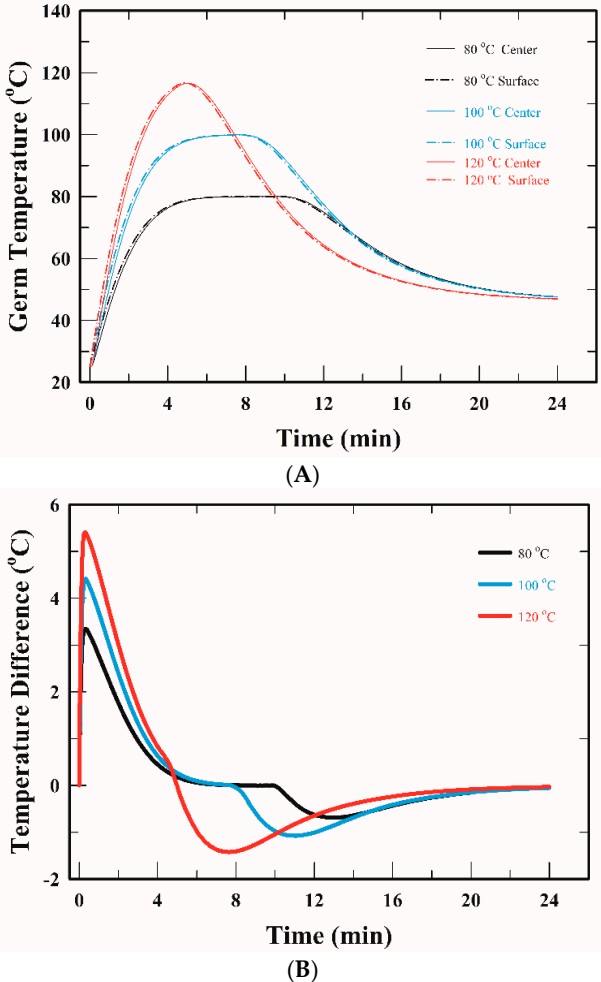

**Figure 9.** Simulated profiles of the temperature distribution (**A**) and the temperature difference (**B**) of WG during drying at different set temperature in the FBD. Solid lines represent the WG center; dash-dot lines represent the WG surface.

### 4.8. Simulated Responses of WG Moisture Distribution

The simulated MC of the WG center and WG surface during drying at different set temperature in the FBD are shown in Figure 10A. The MC of the WG surface was lower than that of the WG center at the heating stage. At the cooling stage, the MC of the WG surface became higher than that of the WG center. This might be due to the condensation of water vapor on the WG surface. The condensation phenomena would occur when the time of heating stage was short [18]. We further divided the changes in the MC within the WG particle during drying into three periods: the decreased period, the dynamic equilibrium period and the increased period.

The MC difference between the WG surface and WG center during drying at different set temperature in the FBD are shown in Figure 10B. The value of the MC difference dropped rapidly at the initial stage of heating and rose to zero with drying time. The value of the MC difference was

positive at the later stage of cooling, indicating that the MC of the WG surface was higher than that of the WG center. This phenomenon was more pronounced at drying at 120 °C. Thus, the condensation on the WG surface during drying in the FBD was strongly related to the set temperature.

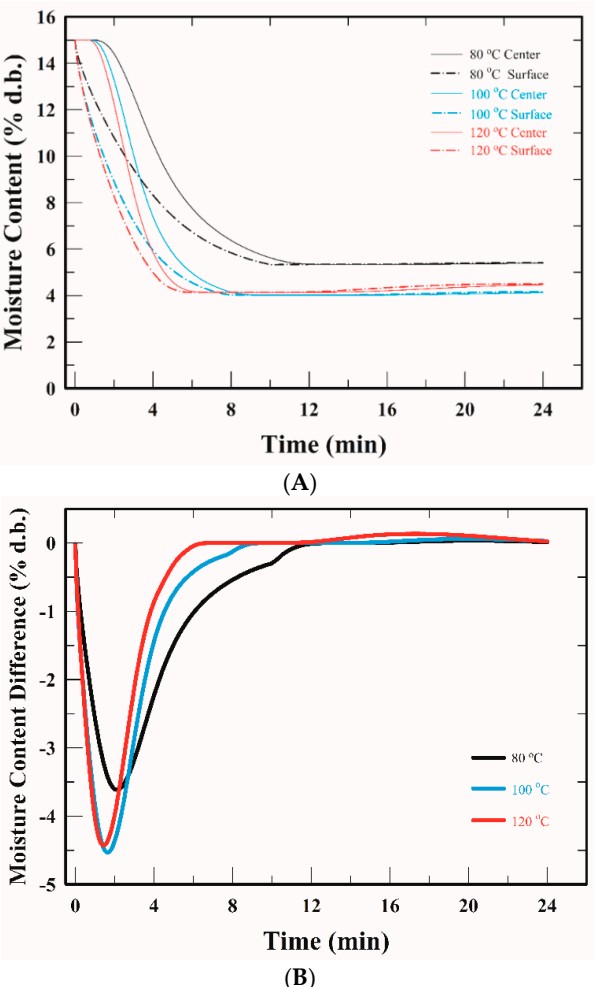

**Figure 10.** Simulated profiles of the moisture content (MC) distribution (**A**) and the MC difference (**B**) of WG during drying at different set temperature in the FBD. Solid lines represent the WG center; dash-dot lines represent the WG surface.

*4.9. Simulated Responses of the Dehydration Flux and the Condensation Flux on the WG Surface*

In order to assess the effects of heating stage and cooling stage on dehydration and condensation during WG drying, the mass flux term of right-hand side in Equation (8) were used. The changes of mass fluxes during WG drying were shown in Figure 11. There are two mass flux peaks during WG drying. The first positive peak was at the heating stage of 0–10 min and the second negative peak was at the cooling stage of 11–24 min. The second flux peak was smaller than the first peak.

For the peaks at the heating stage, the shape of the peak was narrow at the set temperature of 120 °C and it became broader at lower set temperature. The dehydration time could be obtained when the mass flux was zero. The times were 10.1, 8.3 and 6.1 min during drying at 80, 100 and 120 °C, respectively. The heating time was 9.4, 7.5 and 3.9 min at the set temperature of 80, 100 and 120 °C, respectively (Table 1). The prolonging effect at the heating stage was the difference between the dehydration time and the heating time. Therefore, it was 0.7 min (10.1 to 9.4 min), 0.8 min (8.3 to 7.5 min) and 3.1 min (6.1 to 3.0 min) during drying at 80, 100 and 120 °C, respectively. The above results indicated that the WG drying at higher set temperature and shorter heating time (e.g., 120 °C

for 3 min) had stronger prolonging effect. The dehydration of WG at higher set temperature was more efficient than at the lower set temperature.

For the peaks at the cooling stage, the flux peak at the set temperature of 120 °C was broader and higher than the peaks at the lower set temperature, indicating that the condensation of WG was more pronounced at drying at higher set temperature. The condensation time was 9.9 (14.1 to 24.0 min), 10.5 (13.5 to 24.0 min) and 13.8 min (10.2 to 24.0 min) during drying at 80, 100 and 120 °C, respectively. In order to obtain the final product temperature of 45 °C and the WA of 0.3 in this study, a longer condensation time was necessary when WG was dried at higher set temperature. Accordingly, we suggest that a compromise between the efficiency of dehydration step and the efficiency of condensation step needs to be considered for the process optimization.

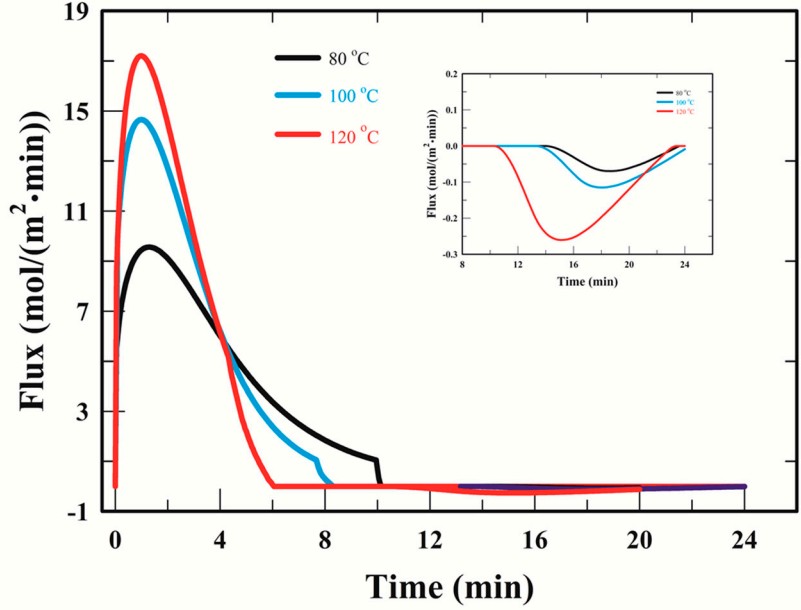

**Figure 11.** Simulated profiles of the mass flux on the WG surface during drying in the FBD at different set temperature. 80 °C (black line), 100 °C (blue line), and 120 °C (red line) in heating stage and cooling stage.

## 5. Conclusions

The performance of WG drying in the FBD was compared for different time-temperature combinations. All combinations could dry the WG to the target temperature of 45 °C and water activity of 0.3. The changes of the MC of WG could be divided into the decreased period, the dynamic equilibrium period and the increased period. A linear relationship between MC (4.0–15.0%) and WA of WG was observed in this study. The WG dehydration at the heating stage and the condensation at the cooling stage could be used to diagnose the drying performance based on the simulated MC of WG center and WG surface. The WG dehydration and condensation could also be evaluated by the dehydration flux and the condensation flux, respectively. According to the results of dehydration flux and condensation flux, the efficiency of dehydration was better at higher set temperature, however, the efficiency of condensation was better at lower set temperature. A process with a higher set temperature (120 °C) and a shorter heating time (3 min) might be recommended for a hot and humid environment such as the summer season in Asia.

**Author Contributions:** D.-S.C. performed the experiments and developed the process model and computational simulations. M.-I.K. was responsible for the conception, research ideas, and research grant.

**Funding:** This research was funded by Texture Maker Enterprise Co., Ltd., Taiwan, grant number 600312.

**Conflicts of Interest:** The authors declare no conflict of interest.

## Nomenclature

| Symbol | Meaning (units) |
|---|---|
| $A_g$ | surface area per unit volume of wheat germ (1/m) |
| $C_{mb}$ | concentration of moisture in bubble phase (mol/m$^3$) |
| $C_{me}$ | concentration of moisture in emulsion phase (mol/m$^3$) |
| $C_{mg}$ | concentration of moisture in wheat germ (mol/m$^3$) |
| $C_{sat}$ | saturation concentration (mol/m$^3$) |
| $C_e$ | equilibrium moisture content (mol/m$^3$) |
| $C_{p,e}$ | effective heat capacity (J/(kg·K)) |
| $D_e$ | effective diffusivity of water in wheat germ (m$^2$/s) |
| $D_o$ | pre-exponential factor (m$^2$/s) |
| $D_v$ | Diffusivity of water vapor (m$^2$/s) |
| $E_a$ | activation energy of water diffusion (J/mol) |
| $f_{con}$ | step numerical parameter for condensation |
| $f_{de}$ | step numerical parameter for dehydration = $(1 - f_{con})$ |
| $H_f$ | height of fluidization (m) |
| $h$ | convective heat transfer coefficient (W/(m·K)) |
| $K_{be}$ | mass transfer coefficient between bubble phase and emulsion phase (1/s) |
| $K_{con}$ | mass transfer coefficient for condensation (m/s) |
| $K_{de}$ | mass transfer coefficient for dehydration (m/s) |
| $M_w$ | molecular weight of water (kg/mol) |
| $P_{sat}$ | saturated vapor pressure of water (Pa) |
| $r_g$ | radius of wheat germ (m) |
| $R_{dl}$ | rate of mass transfer between bubble phase and emulsion phase (mol/(m$^3$·s)) |
| $R_g$ | gas constant (J/(mol·K)) |
| $R_T$ | gas constant (atm/(mol·K)) |
| $RH$ | relative humidity of air (-) |
| $T$ | temperature (K, unless it is specified in °C) |
| $t$ | time (s) |
| $V_T$ | volume of fluidization (m$^3$) |
| $W_L$ | loading of wheat germ (kg) |
| $X_{wi}$ | initial moisture content of wheat germ (% d.b.) |
| $r, x$ | coordination (m) |

**Greek symbols**

| | |
|---|---|
| $\kappa$ | thermal conductivity (W/(m·K)) |
| $\varphi$ | porosity of bed (-) |
| $\rho$ | density (kg/m$^3$) |
| $\lambda$ | latent heat of vaporization (J/mol) |

**Subscript**

| | |
|---|---|
| $b$ | bed |
| $c$ | cooling |
| $con$ | condensation |
| $e$ | effective or emulsion |
| $g$ | germ |
| $h$ | heating |
| $i$ | initial |
| $in$ | inlet |
| $m$ | moisture |
| $mb$ | water vapor in bubble phase |
| $me$ | water vapor in emulsion phase |
| $s$ | set |
| $sat$ | saturation |
| $w$ | water |

## Appendix A

**Table A1.** The experimental and simulated MC, absolute average deviation (AAD) and coefficient of determination ($R^2$) for different set temperature.

| Time (min) | 80 °C | | 100 °C | | 120 °C | |
|---|---|---|---|---|---|---|
| | Exp. | Sim. | Exp. | Sim. | Exp. | Sim. |
| 0 | 15.07 | 15.00 | 15.07 | 15.00 | 15.07 | 15.00 |
| 2 | 10.21 | 11.64 | 9.19 | 9.81 | 8.32 | 9.05 |
| 4 | 8.28 | 8.70 | 6.89 | 6.19 | 5.56 | 5.14 |
| 6 | 7.12 | 6.92 | 5.64 | 4.62 | 4.43 | 4.13 |
| 9 | 5.92 | 5.62 | 4.56 | 4.02 | 3.94 | 4.13 |
| 12 | 5.64 | 5.34 | 4.36 | 4.02 | 3.94 | 4.14 |
| 15 | 5.55 | 5.34 | 4.33 | 4.02 | 3.98 | 4.26 |
| 18 | 5.53 | 5.36 | 4.49 | 4.07 | 4.18 | 4.39 |
| 21 | 5.75 | 5.40 | 4.32 | 4.13 | 4.27 | 4.47 |
| 24 | 5.99 | 5.41 | 4.68 | 4.15 | 4.42 | 4.49 |
| ADD | 0.40 | | 0.47 | | 0.26 | |
| $R^2$ | 0.976 | | 0.989 | | 0.992 | |

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
