# Peer review of "Wheat Germ Drying with Different Time-Temperature Combinations in a Fluidized Bed Dryer"

_processes, doi:10.3390/pr6120245_

Reviewer 1 Report

Introduction

the introduction needs an improvement.

In this paragraph the authors have to improve the advantage and disadvantage of Fluidized bed dryer (FBD) for wheat germ.

Line 30: explain deeply the composition of WG in terms of WA and WC.

line 40: please insert references.

line 44: "The  increase of MC during WG drying at the cooling stage was observed".  please explain.

Materials and methods

table 1: it is not clear that in the table there are minutes. please change.

Results and discussion

lines 203-204: please explain deeply why: "both MC and WA of food are important parameters to assess the stability"

figure 8A: improve the resolution

line 265: please explain this sentence "However, no equilibrium period was observed when the set temperature was 120 °C".

Author Response

Dear Academic editor and Reviewer:

We very appreciate your comments and suggestions. The following is our responses to your comments.

Reviewer 2 Report

The subject of this paper was ‘to investigate the drying performance of wheat germ with different time-temperature combinations in the Fluidized Bed Dryer. In the meantime, a mathematical model was proposed to develop an optimal drying condition’. The experiments involved the measurement of moisture content and water activity, in the process of investigating the drying performance of WG with different time-temperature conditions in the FBD.

The objective and the approach of authors in this manuscript is interesting and the subject studied is of importance for food science. The structure is well laid out, the abstract reflects the key elements of the article content and references cover a wide range of the studies focusing on the particular subject Although I think that the subject is quite appropriate for the scope of the particular journal and the manuscript presents a quite novel process that could be of potential interest to the food industry, I have serious considerations for its publication in the current form (as will be detailed on the next section).

BROAD COMMENTS:

One of the main weaknesses is the poor language, the numerous syntax/grammatical errors throughout the text, that in some cases, make the manuscript incomprehensible and the vague conclusions frequently drawn by the authors without explaining their findings. Another very important comment lies on the degree of similarity with a previous work of this group (Ref. [18] of this manuscript,  Chan, D.S.; Chan, J.S.; Kuo, M.I. Modelling Condensation and Simulation for Wheat Germ Drying in 409 Fluidized Bed Dryer. Processes. 2018, 6(6), 71.), recently published by the same scientific journal. If the current submission is meant to be considered as a follow-up of this previous work, in my opinion, authors should reconsider all their text, in order to avoid all these repetitions. For example, ‘Mathematical modelling part’ (§3) is a copy-paste (exactly in the same words with the relative part of their precious work), not to mention that model validation was already presented in this work (and I do not find the reason to show once more the same model’s validation in the current submission).  I suggest that authors reconsider in depth their submission, avoiding the same details, figures (e.g. Figure 1 and numerous other figures) and deliver a revised text, bearing in mind that the originality of this paper lies on the comparison of FBD performance when different time-temperature regimes are implemented.

Another issue is related to the unnecessarily high number of figures, that should be substantially reduced. Some figures (e.g. Fig 3A provide the same information with Table 3, so the Table could be preferably omitted, if authors want to explain the drying periods, adding the necessary information within the text). Furthermore, there are too many obvious conclusions throughout the text, that affect in a negative way the manuscript originality.

Focusing on the core of the submission, one of the main concerns is related, on one hand, to the selection of the 3 different time-temperature regimes and, on the other hand, to the evaluation of the effect of heating time/temperature, as separate factors. In many cases within the text, authors assess the effect of those distinct operation parameters, which, in my opinion, lacks scientific basis, since they both change in the T-t regimes chosen. To be able to conduct such comparisons and evaluate the effect of each parameter, one should be kept constant (e.g. set temperature) changing the other (drying time). All text should be carefully reconsidered to avoid such statements/conclusions.

Summarizing there are important issues throughout the text that make me reluctant to suggest further publication of this manuscript. In my opinion, in its current form, it needs modifications and serious reconsideration. As far as text improvement is concerned, some minor modifications are suggested to the authors.

SPECIFIC COMMENTS:

Abstract: 

line 20-21: bad expression...please, rephrase

line 22-23: .. bad expression...please, rephrase, since it does not make sense

Introduction

line 33: ..the heating time required...

line 40: ..with a desired light golden-yellow..

line 41: the phrase ‘from a commercial viewpoint’ transferred at the beginning of the sentence.

Line 48: ..proposed to optimize the drying conditions

Materials and methods

§2.4: see also general comments..how were those T-t regimes decided?

Figure 1 the same as in reference [18]

§3: a detailed repetition with [18] with exactly the same words! Authors should reconsider reducing this part with an effective and comprehensive reference to their previous work

Line 86: thermophysical properties

Results and discussion

Line 158: ..were in good agreement with measured data

Line 162: the time reported is the mean time? Values inside parentheses show the range? Please, specify.

Figure 2,4,5 (and also for other figures): explain the dashed line/change legend ‘..set temperatures. Experimental data are represented as symbols; simulated values are represented as lines’

Table 3 and fig 3A provide exactly the same information (see also general comments)

Line 185: ..could be divided (omit ‘about’)

Line 187: Is the statement ‘a slight increased period was obtained at the cooling stage’ correct? Have you checked the statistical difference between exp. Points? Can you sustain this argument from other literature findings and try to further explain it?

Line 190-191: do you mean the rate of decrease or the finally obtained MC value?

How can you really compare those treatments when both time and temperature change simultaneously? I agree with testing the validity and the accuracy of your model (although already tested in [18]), but I question the approach of evaluating the time and temperature (separately) effect on FBP performance.

Line 199: the predicted values (instead of simulated)

Line 203-204: Rephrase..Both MC and WA are important parameters to assess food stability

Line 207: Is this equation valid for all conditions investigated? You provide very few information on the construction and on the goodness of fit for eq. 31. How good was the fitting and how well does this equation work? Indeed, it would be very helpful to be able, through a quick measurement of WA, to predict the MC of the sample! Try to reinforce this (novel) part.

Line 218-219: Isn’t this expected? Avoid obvious conclusions!

Lines 223-230: I do not fully understand the necessity of this analysis (fig 5B). How much different are these findings compared to the ones commented on Fig2 (that demonstrates heater temperature vs. time)? If they present significant differences (I doubt that), how do you explain?

Lines 229-230: I do not agree with that conclusion (see also general comments on the effect of set temperature/heating time, as distinct factors)

Lines 235-240: I do not think that this part is necessary, as I fail to see the meaning if the analysis (the statement is vague and lacks adequate explanation). I also do not fully understand the phrase (line 239): ‘The above results..

Line 243: ..the diffusivity of moisture increased (omitted the word ‘was’)

Line 244: ….80C, as far as the heating stage is concerned

Line 245: ..’WG was related to the WG temperature’..how can you compare when the heating time also varies between the 3 different set temperatures?

Lines 252-260: Not expected? What is exactly the point that authors try to stress out here?

Lines 270-282: This part, in my opinion, provides important findings!

Line 279: ..was more pronounced at drying at 120C

Lines 295-296: Rephrase..not able to understand the meaning

Lines 301-305: Although better explained in the ‘Conclusion’ part, you should include an explanation at this point about the optimized conditions for the whole process (including both heating and cooling). Is a longer condensation time desirable? You could mention that the choice is a compromise between the efficiency of dehydration (higher set temperatures) and that of the condensation step (lower set temperatures).

Conclusions

In this part, you only use findings out of the last part of your study (which is the original and new one). You should try to focus your whole manuscript around this novel work!

Line 311: rephrase ..’in the FBD was compared for different time-temperature combinations’.

Author Response

Dear Academic editor and Reviewer:

We very appreciate your comments and suggestions. The following is our responses to your comments.

Round  2

Reviewer 2 Report

The subject of this paper was ‘to investigate the drying performance of wheat germ with different time-temperature combinations in the Fluidized Bed Dryer. In the meantime, a mathematical model was proposed to develop an optimal drying condition’. The experiments involved the measurement of moisture content and water activity, in the process of investigating the drying performance of WG with different time-temperature conditions in the FBD.

In the revised manuscript, authors incorporated in the text all of the minor suggestions and modified to a significant extent the content of their work, by addressing most of the major issues underlined. Concerning the linguistic errors, the text was reconsidered and significantly improved. The Conclusion part was also significantly modified, to better describe the main findings of this study.

Having still some objections about the evaluation of the effect of heating time/temperature, as separate factors (when both factors time and temperature, are simultaneously changing), there are still some minor issues that should be also addressed to produce a manuscript worthy to be published in Processes Scientific Journal.

Introduction

line 49-50: ..leading to a final product...

lines 60-62: bad language, please rephrase

Mathematical modeling

§3:The incorporation of a new table (Table 2) that describes the calculation of variables is useful.

Results and discussion

Line 192: ..omit the phrase..’was needed’

Figure 2,4,5 (and also for other figures): change legends.. ‘Experimental data are represented as symbols; simulated values are represented as lines’

Line 217: An increase..

Line 222: Rephrase ‘The MC values predicted were in good agreement with the experimental data’

Line 225/227: at the end of the heating stage..’reached a constant value’

Line 255: how good is the fitting of Eq. 26? Provide some values (e.g. R2)

Line 396: Do you  mean necessary (or desirable)?

Line 398: needs to be considered

Author Response

NO

Comments

Responses

1

line 49-50: ..leading to a final product...

This sentence has been revised.

2

lines 60-62: bad language, please rephrase

This sentence has been revised.

3

§3:The incorporation of a new table (Table 2) that describes the calculation of   variables is useful.

4

Line 192: ..omit the phrase..’was needed’

This sentence has been revised.

5

Figure 2,4,5 (and also for other figures): change legends.. ‘Experimental data are   represented as symbols; simulated values are represented as lines’

The information has been revised in the legends.

6

Line 217: An increase..

This sentence has been revised.

7

Line 222: Rephrase ‘The MC values predicted were in good agreement with the   experimental data’

This sentence has been revised.

8

Line 225/227: at the end of the heating stage..’reached a constant value’

This sentence has been revised.

9

Line 255: how good is the fitting of Eq. 26? Provide some values (e.g. R2)

The coefficient of determination (R2 =0.980) has been added in the Figure 5.

10

Line 396: Do you  mean necessary (or desirable)?

This word has been revised.

11

Line 398: needs to be considered

This sentence has been revised.